# Nutraceuticals and Exercise against Muscle Wasting during Cancer Cachexia

**DOI:** 10.3390/cells9122536

**Published:** 2020-11-24

**Authors:** Giorgio Aquila, Andrea David Re Cecconi, Jeffrey J. Brault, Oscar Corli, Rosanna Piccirillo

**Affiliations:** 1Neuroscience Department, Mario Negri Institute for Pharmacological Research IRCCS, 20156 Milan, Italy; giorgio.aquila@marionegri.it (G.A.); andrea.rececconi@marionegri.it (A.D.R.C.); 2Italian Institute for Planetary Health, IIPH, 20156 Milan, Italy; oscar.corli@marionegri.it; 3Indiana Center for Musculoskeletal Health, Department of Anatomy, Cell Biology & Physiology, Indiana University School of Medicine, Indianapolis, IN 46202, USA; jebrault@iu.edu; 4Oncology Department, Mario Negri Institute for Pharmacological Research IRCCS, 20156 Milan, Italy

**Keywords:** cancer cachexia, muscle wasting, muscle atrophy, lifestyle interventions, nutraceutical, exercise, myokine, nutrition, bimodal approach

## Abstract

Cancer cachexia (CC) is a debilitating multifactorial syndrome, involving progressive deterioration and functional impairment of skeletal muscles. It affects about 80% of patients with advanced cancer and causes premature death. No causal therapy is available against CC. In the last few decades, our understanding of the mechanisms contributing to muscle wasting during cancer has markedly increased. Both inflammation and oxidative stress (OS) alter anabolic and catabolic signaling pathways mostly culminating with muscle depletion. Several preclinical studies have emphasized the beneficial roles of several classes of nutraceuticals and modes of physical exercise, but their efficacy in CC patients remains scant. The route of nutraceutical administration is critical to increase its bioavailability and achieve the desired anti-cachexia effects. Accumulating evidence suggests that a single therapy may not be enough, and a bimodal intervention (nutraceuticals plus exercise) may be a more effective treatment for CC. This review focuses on the current state of the field on the role of inflammation and OS in the pathogenesis of muscle atrophy during CC, and how nutraceuticals and physical activity may act synergistically to limit muscle wasting and dysfunction.

## 1. Introduction

### Definition and Classification of Patients with Cancer Cachexia

Cancer-induced cachexia (CC), a debilitating syndrome characterized by progressive loss of skeletal muscle mass and function (with or without fat loss), affects about 80% of patients with advanced cancer [1,2]. Besides peripheral muscle atrophy, patients with CC also experience cardiac atrophy, remodeling, and dysfunction that can lead to heart failure [3,4], as well as diaphragm atrophy that can cause respiratory collapse [5]. Both conditions contribute to poor quality of life and reduced survival in 30–40% of cases. CC is still an unmet medical problem.

Three progressive stages of CC have been classified in [2], based on disease severity: pre-cachexia, cachexia, and refractory cachexia. The goal of this classification is to identify patients who are likely to benefit from early interventions and to limit and/or reverse body weight and muscle loss. Pre-cachexia involves only metabolic changes, such as impaired glucose tolerance, causing negligible body weight loss (BWL). Cachexia is defined by BWL > 5% in six months or BWL > 2% and body mass index (BMI) < 20 kg/m^2^ or muscle depletion and more severe metabolic derangements. Refractory cachexia is distinguished by unresponsiveness to cancer treatment where its management is no longer possible [2]. Life expectancy decreases through the various stages of CC, from 6–9 months for patients with pre-cachexia to less than 3 months in individuals with refractory cachexia. Human body composition (i.e., fat and muscle content) are usually estimated by computed tomography or more indirectly by bioelectrical impedance analysis [6]. Additional diagnostic parameters, such as anemia, anorexia, loss of muscle strength [7], and possible genetic propensities, can further complicate the classification and the progression of cachectic cancer patients.

In 2017, among the guidelines described by the European Society for Clinical Nutrition and Metabolism (ESPEN) for nutritional care in cancer patients (reviewed in [8]), the following key recommendation was highlighted: “Use nutritional intervention with individualized plans, including care focused on increasing nutritional intake, decreasing inflammation and hypermetabolic stress, and increasing physical activity”. This “call-to-action” by ESPEN experts emphasized the importance of both nutritional interventions—including food-derived products (i.e., nutraceutical compounds)—and exercise to counteract CC. While the beneficial effects of physical activity on survival [9], as well as the anticancer effects of nutraceuticals are known [10], whether and how physical activity and nutraceuticals together prevent or circumvent CC is still unclear. According to the American Society of Clinical Oncology (ASCO) guidelines, “outside the context of a clinical trial, no recommendation can be made for other interventions, such as exercise and nutritional supplements, for the management of CC” [11].

Our main goal here is to review the role of nutrition, nutraceuticals, and exercise in the treatment of muscle wasting associated with cancer. First, we discuss the pathogenic mechanisms linked to muscle atrophy during CC, emphasizing the contributing role of oxidative stress (OS). Next, we highlight how changes in nutritional habits, nutraceutical interventions, and physical activity can help to preserve muscle mass and function, providing evidence from preclinical models and cachectic patients. Last, we look ahead to future investigations in patients, such as the evaluation of their redox status, to intervene effectively to improve their quality of life.

## 2. Pathogenesis of Cancer Cachexia: Inflammation and Oxidative Stress

### 2.1. Mechanisms of Muscle Wasting During Cancer Cachexia

Cancer cachexia is accompanied by an increase in systemic inflammation [12]. The inflammation causes systemic dysmetabolism in the host where the tumor growth deprives the body of amino acids, nutrients, and energy fuels, at the expense of organs such as skeletal muscles [13]. Muscles are comprised of a combination of glycolytic myofibers, mainly relying on glucose and glycolytic metabolism, which are able to exert strong force in a short time, and oxidative fibers that are more resistant to fatigue because of a richer concentration of ATP-producing organelles, such as mitochondria, and greater vascularization for enhanced oxygen delivery. Oxidative fibers are more resistant to atrophy subsequent to fasting or cancer-related inflammation, but they undergo atrophy more rapidly due to disuse [14]. It is estimated that muscle catabolism increases by 40–60% in humans during cachexia [15]. Most tumors are highly metabolically active and dependent on glucose and glutamine for their growth [13]. Therefore, tumor growth deprives the muscles of nutrients and causes a shift in muscle metabolism, which, converting its structural proteins into amino acids, in turn fuels the liver to produce glucose (gluconeogenesis) as a survival plan and to mount the hepatic acute phase response to inflammation further consuming energy.

Pro-inflammatory molecules, such as interleukin-6 (IL-6), tumor necrosis factor alpha (TNFα), IL-1, and interferon-γ, derived from the immune system or from the tumor itself and glucocorticoids, are increased in plasma of rodents and patients with different cancers [16]. Their induction is not sufficient to induce cachexia, as some individuals or animal models have increased levels of these inflammatory molecules and do not present muscle wasting [17,18]. Further, at the cellular level, myotubes in vitro exposed to high concentrations of these cytokines do not consistently present increased proteolysis [19], confirming that other factors alone or in combination with inflammatory cytokines are more likely to cause muscle wasting directly [20].

In clinical trials, the neutralization of circulating IL-6 or TNFα have not proven useful to counteract CC, indicating that other factors are involved in cachexia [21,22]. Indeed, novel molecules have recently been identified as triggers of muscle wasting during cancer, at least at the preclinical level, such as leukemia inhibitory factor (LIF) [23], TNF-related weak inducer of apoptosis (TWEAK) [24], the negative regulator of muscle mass myostatin [25,26], High Mobility Group Box 1 (HMGB1) [13], tumor-derived parathyroid hormone-related protein [27], growth differentiation factor 15 (GDF-15) [28], and microRNAs from the tumor itself [29,30,31]. Regrettably, the concomitant inhibition of many of them is unpractical at this time.

In 2004, a set of genes were identified that were differently expressed in skeletal muscles from rodents undergoing atrophy due to a variety of conditions (disuse, uremia, cancer, and diabetes); these genes were defined as “atrogenes” [32]. Atrogenes comprise upregulated genes encoding for many subunits of the 26S proteasome, ubiquitin ligases such as atrogin-1 and Muscle RING-finger protein-1 (MuRF1), and ubiquitin—all indicative of the enhanced protein degradation common to all atrophying muscles analyzed. Atrogenes that were reduced include certain growth-associated proteins such as the peroxisome proliferator-activated receptor γ coactivator 1α (PGC1α) and JunB.

In skeletal muscles, transcription factors such as forkhead box O3 (FoxO3), also belonging to the family of atrogenes, nuclear factor kappa-light-chain-enhancer of activated B cells (NF-kB), signal transducer and activator of transcription 3 (STAT3) are also all activated by tumor-related inflammation. They enhance the expression of genes encoding molecules, such as atrogin-1 and MuRF1, that promote the proteasomal degradation of muscle proteins, leading to atrophy [33,34,35,36]. In detail, MuRF1 (also known as TRIM63) is responsible for the degradation of myosin heavy chain [37] and other components of the thick but not the thin filaments [38]. Instead, atrogin-1 (known also as MAFbx or Fbxo32) ubiquitinates and so promotes the proteasomal digestion of MyoD (a transcription factor crucial for muscle differentiation) [39]. Both atrogin-1 and MuRF1 are induced in cachectic muscles of cancer-bearing rodents [32] and in the muscles of cancer patients even before muscle depletion [40]. Other less studied ubiquitin ligases involved in the accelerated proteolysis typical of CC include TNF receptor-associated factor 6 (TRAF6) [41,42,43], the ubiquitin ligase E3α-II [44], and muscle ubiquitin ligase of Skp, Cullin, F-box (SCF) containing complex in atrophy-1 (MUSA1) [45].

During muscle wasting, FoxO3 induces a transcriptional response that not only promotes the proteasomal degradation of proteins but also coordinately induces genes involved in the degradation of proteins and organelles through the autophagic pathway [46,47]. Destruction of proteins from the sarcomere contributes to loss of muscle strength, while removal of ATP-producing organelles, such as mitochondria by autophagy (mitophagy), contributes to the lower resistance to fatigue typical of cancer patients. Another player coordinately involved in diverting proteins to proteasomes and lysosomes in skeletal muscle is the p97/Valosin-containing protein (VCP) ATPase complex [48,49]. Unexpectedly, its expression is not under the control of FoxO3 [50] but another transcription factor named Pax4 [51].

The main signaling pathway leading to increased protein synthesis involves AKT/phosphatidylinositol-3-kinase (PI3K) and results in the activation of eukaryotic translation initiation factor 4E-binding protein (4E-BP1) and ribosomal protein S6 kinase beta 1 (S6K1), which, together, enhance protein translation. Major activators of this pathway are insulin-like growth factor 1 (IGF1) and insulin. Conversely, the growth-promoting transcription factor JunB exerts its hypertrophic effects in an AKT-independent way, reducing atrogene expression overall [52]. These pathways are depressed in muscles undergoing cancer cachexia [53,54]. As a consequence, not only is protein degradation increased but protein synthesis too seems to be attenuated, further exacerbating protein loss, as in murine colon adenocarcinoma 16 (MAC16)-bearing rodents [55], Lewis Lung carcinoma (LLC)-carrying mice [56], and in cancer patients [57]. Interestingly, atrogin-1 also restrains protein synthesis by promoting the degradation of the elongation factor eIF3-f [58], further aggravating muscle loss during wasting. Similarly, by ubiquitinating and degrading the insulin receptor substrate 1 (IRS1) that is an important signaling interactor of insulin/IGF1 receptor, the ubiquitin ligase Fbxo40 is involved in restraining IGF1-mediated protein synthesis [59].

Muscle wasting due to net protein loss, with no reduction in the number of myofibers (i.e., hypoplasia), is reversible. Removal of the primary tumor before appearance of metastases—at least in the early stages (i.e., before cachexia becomes refractory)—is sufficient to reverse the initial muscle atrophy in mice [60] and in rats [61]. Sometimes removal of the primary tumor is impossible, and the patient goes through progression of disease and development of metastases. Metastasis and cachexia should be thought of in parallel, highlighting the overlapping pathways between them [62]. Inflammation and metabolic dysregulation are key factors in the pathophysiology of cachexia and at the same time of metastases and tumor progression [63]. In cancer patients as in old individuals, a complicating factor has been reported: anabolic resistance. This may explain the third stage of the disease, refractory cachexia. Molecular adaptations to tumor burden in skeletal muscles result in refractoriness to the anabolic signaling pathways (i.e., AKT, mammalian target of rapamycin (mTOR), JunB) usually activated by nutrition and growth-stimulating exercise (anaerobic physical activity). This has also been observed in preclinical animal models [64] as well as in advanced cancer patients suffering from cachexia [65] and currently indicates a point of no return in CC management.

Strategies aimed at preserving both contractile proteins and mitochondria should be pursued to preserve muscle strength and alleviate muscle weakness in patients at the same time. Activation of PGC1α through aerobic exercise could be useful to exert both these protective effects. PGC1α is the key transcriptional co-activator promoting mitochondrial biogenesis [66] and an antioxidant defense (mainly increasing nuclear factor erythroid 2-related factor 2 or Nrf-2) [67]. At the same time, it antagonizes the DNA binding of FoxO3, impairing atrogene transcription, avoiding the concomitant increase in proteasomal and autophagy-mediated proteolysis [68], and ultimately protecting mice from muscle atrophy [69]. PGC1α also controls the expression of muscle-secreted factors (i.e., myokines), such as irisin or musclin [70,71]. We recently found the PGC1α-musclin axis was depressed in atrophying muscles during cancer [70] and that restoring the expression only of musclin in cachectic muscles of cancer-bearing mice protects them from fiber atrophy [70].

### 2.2. Link between Oxidative Stress and Muscle Wasting

To further complicate the known transcriptional mechanisms of protein loss, OS is thought to be a key contributor in the deleterious process of muscle wasting in cancer or other chronic diseases. A commonly accepted definition of OS is “an imbalance between oxidants and antioxidants in favor of the oxidants, leading to disruption of redox signaling and/or molecular damage” [72]. Oxidants are those substances that can generate reactive oxygen species (ROS) or induce OS, while antioxidants include substances that when at low concentrations with respect to a certain substrate are enough to delay or prevent its oxidation [73]. ROS, comprising both free radicals, such as superoxide anion (O_2_^•−^), and nonradicals, such as hydrogen peroxide (H_2_O_2_), are part of a bigger family referred to as reactive species that encompasses reactive nitrogen and reactive chlorine species. Since O_2_^•−^ is the primary free radical that gives rise to major cell-damaging reactants (e.g., hydroxyl free radical, OH^•^) [72], most of this review will refer to ROS in CC.

In mammalian cells, O_2_^•−^ originates from NADPH oxidases (NOX), the mitochondrial electron-transport chain, xanthine oxidases, cyclooxygenases and lipoxygenases, uncoupled nitric oxide synthases (NOS), and cytochrome P450s. By contrast, the antioxidant defense depends on enzymes such as superoxide dismutase (SOD), catalase, and glutathione peroxidase (GPx) or non-enzymatic molecules such as vitamin C, vitamin E, glutathione, and beta-carotene. Transferrin, ceruloplasmin, and albumin also act as indirect antioxidants by buffering metal ions such as iron and copper, preventing them from reacting with H_2_O_2_ and ultimately avoiding the generation of the toxic OH^•^ [74]. The less recognized protein metallothioneins exert similar buffering and protective role [75].

Among the atrogenes originally identified [32], activating transcription factor 4 (ATF4) [76] and nuclear factor, erythroid derived 2, like 2 (Nfe2l2, also named Nrf-2) [77], are transcription factors that promote the expression of genes controlled by antioxidant response elements. They were both induced 3–4-fold in all four states tested, including muscles atrophying because of cancer [32]. This indicates that inducing both the atrogenes ATF4 and Nfe2l2 are part of a means to regulate ROS. Notably, preventing FoxO3 action results in loss of induction of at least ATF4 during atrophy [78]. On the other hand, the notion that enhanced OS contributes to muscle wasting is also supported by SOD1 inactivating mutations that lead, among other effects, to muscle depletion in animal models and patients with amyotrophic lateral sclerosis (ALS) [79,80]. When a cachexia-inducing tumor such as LLC was injected in SOD1 knock out (KO) mice, the cancer-induced muscle loss and dysfunction were not further exacerbated by OS, while a quarter of the mice died earlier than LLC-bearing WT mice [81]. How SOD1 depletion affects premature death in LLC-carrying mice is still not clear.

Even more convincing was the drastic induction of the expression of metallothioneins in all atrophy conditions tested [32]: they were among the most induced atrogenes in atrophying muscles (3- to 20-fold, depending on the isoform). Metallothioneins are low-molecular-weight cysteine-rich zinc-binding proteins that are induced by heavy metals, glucocorticoids, and OS; they can protect cells against DNA damage from ROS and detoxify cells from heavy metals [75]. It was recently reported that ablation of metallothioneins 1 and 2 in myotubes and in adult muscles preserves mass and strength in glucocorticoid-induced atrophy, making the beneficial role of their induction during atrophy debatable [82]. To further complicate the picture is the fact that heavy metals, such as zinc, are increased at the onset of muscle wasting during cancer, at least in preclinical animal models [83].

Despite the concomitant activation of these detoxifying systems (ATF4, Nfe2l2, metallothioneins) in emaciated muscles, OS seems not to be resolved in muscles during wasting in general and during CC in particular. Concomitant induction of prooxidant species and antioxidant players, ultimately resulting in increased OS, has been seen in muscles of AH-130 hepatoma-bearing rats and C26 colon adenocarcinoma-carrying mice [84,85]. In transforming growth factor beta (TGF-β)- [86], TNFα-induced muscle atrophy [87], or even in animal models of CC [88] or genetically modified animals displaying muscle wasting (i.e., skeletal muscle-specific transforming growth factor-β activated kinase 1 or Tak1-KO mice [89]), increased ROS levels divert muscle fibers towards enhanced protein degradation, leading to muscle loss [90]. Notably, angiotensin II has been shown to increase ROS-induced proteasomal pathway in muscles of MAC16-bearing mice [91], as in other diseases associated with muscle wasting [92]. Among the possible molecular mechanisms, it seems that increased circulating TNFα typical of CC can trigger muscle OS and increase NOS, impairing the expression and DNA binding ability of myosin creatinine phosphokinase, reducing the related expression of contractile proteins, ultimately causing atrophy [88]. In addition, the reduction in expression and activity of antioxidant enzymes, such as GPx, despite the induced SOD1 and the reduced NOX, ultimately results in the increase in superoxide content, contributing to CC in muscles of MAC16-bearing mice [93] and seemingly in patients suffering from gastric or esophageal or pancreatic cancers [94].

Molecules signaling towards muscle wasting that can sense intracellular ROS are NF-kB [95], further linking inflammation with OS, PI3K/AKT/mTOR [96] and FoxO3 [97]. A summarized scheme showing how OS impacts on muscle anabolic and catabolic pathways is shown in Figure 1. During CC, ROS can promote muscle protein degradation through the ubiquitin proteasome system (UPS) stimulation [90], but also augmented nitrosative stress that occurs subsequently to uncontrolled nitric oxide accumulation plays a role in the hypercatabolism during CC [98]. In particular, protein carbonylation—measured by total levels of carbonyl group formation and both hydroxy-4-nonenal-(HNE) and malondialdehyde (MDA)-protein adducts—as well as protein tyrosine nitration are aberrantly high in muscles of cancer-bearing rats with late-stage cachexia, but levels of antioxidant enzymes such as Mn-SOD, catalase, and heme oxygenase-1 (i.e., HO-1) did not change in muscles [98]. To test the effects of antioxidants such as α-lipoic acid, N-acetylcysteine (NAC), and amifostine, ROS levels and antioxidant enzymes, such as GPx and SOD, have been measured in blood of patients with cachexia and ROS found to be increased, while GPx and SOD decreased, overall resulting in OS increase [99].

The scenario is complicated even more by the ability of inflammatory molecules that are increased in the blood during CC to bypass the blood–brain barrier and promote neuroinflammation [100,101,102,103]. This in turn stimulates hypothalamic serotonergic activity and promotes tryptophan degradation into free radicals via the kynurenine pathway, thus favoring OS [104]. Overall, this mechanism seems to contribute to loss of appetite and then anorexia during CC [105]. Notably, IL-1 appears to be more potent in inducing anorexia than IL-6 [106]. A bed-ridden state and loss of appetite further aggravates the muscle loss of cancer patients, setting up a vicious cycle that is hard to break so as ultimately to preserve the individual’s health. These aspects are consistent with the symptomatologic picture of CC patients where asthenia and anorexia are often pronounced. Overcoming this central brake to food intake in these patients is another challenge in the development of efficient anti-cachexia therapies.

Finally, it seems that inflammation and OS are intimately connected, and one can exaggerate the other (Figure 1), giving rise to various disease states [74]. Consequently, therapies based only on anti-inflammatory or antioxidant compounds are likely to fail. Ways to counteract and/or limit OS have been tested to restrain muscle wasting during cancer progression. Some nutraceuticals listed further on are especially useful to reach this aim and keep muscles healthy in cancer patients, whose metabolism is widely dysregulated by cancer progression.

## 3. From Nutrition to Nutraceuticals against Cancer Cachexia

### 3.1. Cachexia Cannot Be Fully Reversed by Nutritional Supplementation

“Let food be your medicine and medicine be your food”: this was the aphorism that Hippocrates, the Greek father of medicine, formulated over 2000 years ago to sum the idea of using nutrition to promote good health and to treat pathologic conditions. Malnutrition is a common complication in cancer patients and a significant contributor to morbidity and mortality in malignancies [107,108]. How to manage malnutrition in cancer patients is still controversial. While some studies support personalized nutrition and programs of nutritional counseling for cancer patients [108,109,110,111], others report that increasing nutritional intake is not enough to counteract CC [2,7,108] and reduce mortality or secondary cancers [112,113,114,115]. On the other hand, CC patients subjected to aggressive refeeding can develop an overfeeding reaction during the first 2–3 weeks [116]. This can be lethal and consists of severe electrolyte and fluid shifts, with aberrant glucose metabolism, hypophosphatemia, hypomagnesaemia, and hypokalemia, due to metabolic derangements [117]. Introduction of calories must be slow and progressive (20 kcal/kg per day initially) in CC patients, under close medical control to avoid undesirable and possibly dangerous reactions.

The cachexia that cancer patients often develop is a multifactorial syndrome characterized by loss of appetite and weight, leading to fatigue and functional impairment, increased treatment-related toxicity, poor quality of life, and reduced survival [11]. Low food intake is an important component of weight loss, but an aberrant metabolism, amply described in this review, is also implicated in its genesis [11]. Symptoms such as pain, dysgeusia, nausea, constipation, and depression can also contribute appreciably to poor food intake [118]. Unlike starvation, that is linked to environmental nutrient scarcity or voluntary choice not to feed and leads to an increased fat and protein catabolism, cachexia is characterized by an appetite suppression and reduced ability to feed, with strong negative metabolism of all fat, protein, and basal metabolic rate [119]. While starvation induces a similar reduction in both energy intake and expenditure, cachexia promotes decreased energy intake and increased expenditure. Overall, cachexia causes metabolic adaptations that diverge from both starvation and malnutrition and depletes the body energy stores, ultimately leading to death [119]. For these reasons, cachexia cannot be fully reversed by nutritional supplementation. Consistently, even phase III clinical trials testing hormone-based therapies (i.e., anamorelin) in the attempt to increase appetite in patients afflicted by non-small-cell lung cancer cachexia failed to ultimately preserve their muscle strength (ClinicalTrials.gov Identifiers: NCT01387269 and NCT01387282).

The usual diet, without attention to the quantity and quality of the food taken, fails to prevent the onset and development of CC. The dietary advice had to specify the type and amount of food, the eating frequency, the calories and protein amounts to achieve daily, and the dietary restrictions. Nevertheless, a recent study on an animal model [120] focused on the distinct metabolic substrates evaluating the impact of two types of nutrients, sugar and lipids. The authors compared the influence of oral intake of glucose (0, 10, 50% solutions) and 2% lauric acid, a medium-chain fatty acid (MCFA), on skeletal muscle atrophy and tumor growth. Additional glucose intake had no effect on muscle loss but promoted tumor growth. MCFA prevented the loss of skeletal muscle mass and suppressed tumor growth. The combination of glucose and MCFA together was better at protecting against tumor-induced muscle mass loss but allowed the tumor to grow further than with MCFA alone. Therefore, the combined intake of these different macronutrients may help alleviate CC and is expected to be proposed for future clinical applications.

A further contribution to assessing the role of diet on the mechanisms of OS involved in the genesis and progression of cancer cachexia comes from a recent review [121], which highlights how some dietary components reduce or exacerbate inflammation and OS. High glycemic load determines hyperinsulinemia, which is linked to cell proliferation in various cancers [122]. Excessive animal protein intake increases ROS production and promotes antioxidant instability. Similarly, a high-fat diet promotes the production of ROS, along with the elevation of TNFα, resulting in chronic inflammation [123]. A common denominator of these observations is the excess intake of the nutrients.

Selected dietary components can affect cancer development. The protective effect of several vitamins in prostate cancer has been widely demonstrated in clinical trials and laboratory experiments [124]. Vitamin E has been reported to reduce the oxidative damage and help stabilize disease progression, the clinical picture, and median progression-free survival [125]. Vitamins A and D too give a positive response in prostate cancer. Among minerals, usually absorbed from drinking water and food, selenium in particular has antioxidant properties, leading to a reduction in ROS [126]. The authors conclude that a diet high in animal proteins and carbohydrates and excessive fat consumption can generate ROS, resulting in OS with all the consequences on tumor development and cachexia. The final dietary advice includes reducing carbohydrates, consuming moderate amounts of calories, reducing overcooked meats, saturated and total fats, replacing refined carbohydrates with whole grains, and increasing vegetable and fruit intake.

The general role of nutrition in CC patients has been the topic of several recent reviews. A 2012 meta-analysis on dietary advice to adult cancer patients indicated that oral nutritional interventions did not affect mortality but could improve the quality of life [127]. In a 2014 systematic review focused on dietary counseling in patients with advanced cancer and cachexia, the authors found that the moderate quality of the studies included did not allow any firm conclusion on the effectiveness of nutritional interventions [128]. Instead, a more recent meta-analysis from 2018 [129] stated that dietary counseling and/or oral nutritional supplements were associated with improved body weight in cancer patients receiving chemo/radiotherapy.

Finally, in 2020 the ASCO published a systematic review from which some recommendations were made on the clinical management of cachexia in adults patients with advanced cancer [11]. The panel of experts moderately recommended in favor of dietary counseling, to produce some benefits that need to be complemented by medications or other strategies.

### 3.2. Nutraceuticals

In tackling the nutrition-related issues in CC, pre-clinical and clinical research is shifting towards promising nutraceutical-based schemes. The neologism “nutraceutical” was coined in 1989 by joining the words “nutrition” and “pharmaceutical” by Dr. Stephen De Felice to refer to substances that are “a food (or part of a food) that provides medical or health benefits, including the prevention and/or treatment of a disease” [130]. Nutraceuticals include “functional foods”, which are whole, or “fortified, enriched and enhanced” foods that supply the required amount of essential nutrients (e.g., vitamins, fats, and minerals) to confer health benefits and, most importantly, “dietary supplements” [131]. These latter, as indicated by the Food and Drug Administration (FDA) agency, are “products taken by mouth (as pills, capsules, tablets, or liquid) that contain a dietary ingredient”. Dietary ingredients include vitamins, minerals, herbs, or botanicals, as well as other substances that can be used to supplement the diet.

Nutraceuticals, particularly dietary supplements, cannot replace conventional medicine since, accordingly to the FDA agency, they “are not intended to treat, diagnose, cure, or alleviate the effects of diseases”. Nonetheless, some of them provide a promising source of compounds helpful in reducing the risk and/or progression of some widespread diseases, such as atherosclerosis [132] and cancer [133]. In recent years, several classes of food- and plant-derived nutraceuticals have also shown the potential for limiting CC, and thus for improving patients’ quality of life. Those for which the major experimental evidence is available are discussed below, and their way of action at the molecular level is summarized in Figure 2.

#### 3.2.1. Omega-3 Fatty Acids

Omega-3 (n-3) fatty acids are polyunsaturated fatty acids (PUFAs) that contain the first carbon–carbon double bond in the third position from the omega (methyl) end of the molecule. PUFAs, and in particular eicosapentaenoic acid (EPA) and docosahexaenoic acid (DHA), are mainly sourced from marine fish and fish oil. They offer various health benefits, most likely due to the resolution of inflammation, throughout the production of resolving lipid mediators, such as resolvins, maresins, or protectins (reviewed in [134]). Both EPA and DHA partly restrain inflammation by limiting the production of highly inflammatory eicosanoids, such as prostaglandins and leukotrienes, derived from the n-6 fatty acid arachidonic acid (AA) [135]. PUFAs also interfere with NF-kB-mediated inflammation through the inhibition of Toll-like receptor 4 (TLR4) [136] and activation of peroxisome proliferator-activated receptor gamma (PPARγ) signaling [137].

Other mechanisms include PUFA binding with G-protein-coupled receptor 120 (GPR120) [138], leading to strong anti-inflammatory and insulin-sensitizing effects and modulation of microRNA expression towards a less inflammatory profile [139]. Inhibiting prostaglandins, TLR4, and NF-kB have all proven useful to counteract muscle wasting during cancer [33,140,141,142], so a compound that attenuates all of them at once should be even more powerful against CC.

The dissection of the possible mechanisms behind the protective effects of PUFAs against CC is the subject of several recent reviews [143,144,145,146,147], and the common thread is the acute phase response reduction, with lower serum levels of C-reactive protein (CRP), TNFα, and IL-6, as the main effect underlying the CC prevention. In C2C12 myoblasts, EPA abolished protein degradation induced by the proteolysis-inducing factor (PIF) by blocking AA release from membrane phospholipids, thus attenuating its conversion to prostaglandins E2 and F2α and to 5-, 12-, and 15-hydroxyeicosatetraenoic acids [148]. In C2C12 myotubes, PUFAs not only inhibited protein degradation by acting on the PPARγ/NF-kB pathway, but also increased the rate of protein synthesis [149,150,151]. Moreover, both PUFAs and their lipid peroxidation products (4-hydroxyhexenal, HHE and HNE) have anti-cachexia effects by preventing the attenuation of myosin expression and myotube formation in C2C12 cells exposed to medium conditioned with human lung tumor cells [152].

In MAC16-bearing mice, EPA antagonized CC-related muscle wasting through down-regulation of proteasome subunit expression and suppression of protein catabolism [153]. In their muscles, EPA also inhibited the rise in pro-inflammatory prostaglandin E2 in response to tumor-produced catabolic factors [154]. In Walker 256 tumor-bearing rats, supplementation of Oro Inca Oil, rich in α-linolenic fatty acid (ALA), reduced the circulating levels of IL-6 and TNFα and preserved body mass. However, it cannot be excluded that this effect was indirect because of the primary anticancer effect exerted by ALA, as indicated by reduced tumor growth [155]. Conversely, in C26-bearing mice, PUFAs improved neither muscles nor BWL but reversed the increase in size of livers and spleens [156], both signs of CC, as with anti-cachexia drugs at preclinical level [157,158]. On the other hand, a combination of nutritional supplements, which includes PUFAs, seems to reduce CC-related inflammation better and improve the functional performance in C26-bearing mice [159,160].

Similarly, even if some studies indicate that PUFAs may improve body weight, lean body mass, quality of life, and overall survival of patients with cancers, especially those with colorectal malignancies [147] by restraining inflammation [161], others report poor or no significant effect of PUFA diet supplementation on these same clinical parameters in patients with CC [162]. These discrepancies may be due to the large range of fish oil dosages and methods adopted (as discussed in [163]), as well as to the lack of knowledge about both the inflammatory and OS status of recruited patients, which might have helped identify those more responsive to PUFAs. Therefore, PUFA against CC is still under debate and the ongoing clinical trial “Enteral Omega 3 During Radiotherapy to Improve the Quality of Life and Functionality of Head and Neck Cancer Patients” (NCT03720158) might help clarify this issue.

However, PUFAs appear to have both antioxidant and prooxidant activities in a context-dependent manner. For instance, both EPA and DHA attenuate OS-induced damage through upregulation of the Nrf-2/HO-1 axis in vitro, in vascular endothelial cells and in adipocytes treated with H_2_O_2_ [164,165], and in vivo in models of brain ischemia [166] and cardiac ischemia/reperfusion injury [167]. Conversely, beneficial effects of PUFAs against malignancies seem to rely on PUFA-driven apoptosis in cancer cells by downregulating NF-κB [168] and increasing lipid peroxidation products to enhance cellular OS [135,169].

The PUFAs seem to counteract the hypermetabolism—hence the hypercatabolism—of skeletal muscles after severe burn trauma, which also exacerbates oxidative and mitochondrial stress [170]. Then too, in pancreatic cancer patients, dietary PUFA supplementation reduced the resting energy expenditure, whose elevation is indicative of hypermetabolism, ultimately increasing overall survival [171]. Similarly, since hyperlipidemia and the subsequent excessive fatty-acid-induced OS promote muscle atrophy during CC [172], the hypolipidemic effects of PUFAs, already highlighted at the end of the 1980s [173], may be beneficial and may also include a regulated expression of the uncoupling proteins (UCP). These are members of the mitochondrial carrier family involved in the control of ROS production, generated by the mitochondrial electron transport chain. Muscles of cachectic rodents show increased expression of both UCP-2 and UCP-3 [174,175,176], the latter also increasing in atrophying muscles of cancer patients [177]. Moreover, skeletal muscle UCP-3 expression is increased in pathologic conditions such as diabetes and starvation [178], with a concomitant increase in circulating free fatty acids (FFAs).

FFA levels are also high in cachectic patients [179], and it is known—at least in patients with hyperlipidemia—that PUFAs lower these levels [180]. Understanding whether hypolipidemic agents, like PUFAs, by means of the reduced circulating FFAs, can abolish UCP-3 expression in muscles, and thus the rate of OS, calls for further exploration.

#### 3.2.2. Natural Polyphenols

Natural polyphenols comprise a group of heterogeneous organic molecules found in various plants and their derivatives, especially fruit and vegetables, but also herbs, cocoa, and tea. Polyphenols contain multiple phenol units and are classified based on the number and oxidation status of these primary units, as well as the presence of other functional groups. Numerous studies already support their potential health benefits, as modulators of OS, for managing and treating chronic diseases, including cardiovascular diseases and cancers [181,182,183]. Supporting evidence also exists for natural polyphenols to limit or restrain CC.

Epigallocatechin-3-gallate (EGCG), belonging to the flavonoid family (a class of plant-derived polyphenolic secondary metabolites), has anti-cachexia effects in vitro and in vivo. In differentiated murine myotubes subjected to serum starvation or exposed to TNFα, 10 μM EGCG rescues protein synthesis and reduces the rate of protein degradation [184], protecting the cells from atrophy. In vivo, EGCG preserves skeletal muscle activity in dexamethasone-treated Wistar rats by increasing muscle acetylcholine sensitivity [185], while in LLC-bearing mice, it attenuated skeletal muscle atrophy by restraining NF-kB expression and downstream atrogenes [186]. Depending on the dose used, EGCG is a possible adjuvant in cancer therapy, because it can exert both antioxidant (low doses) and pro-oxidant (high doses) activities, thus preventing cancer onset or inducing cancer cell death, respectively [187]. For instance, daily intake of six tablets of green tea polyphenols (equivalent to 474 mg/day of EGCG) attenuated ROS levels in plasma of patients with metastatic liver cancers undergoing hepatic arterial infusion chemotherapy [188].

Similarly to EGCG, curcumin, the main polyphenol component of curcuma extract, shows anti-inflammatory, antioxidant, antiatherosclerotic and neuroprotective activities [189]. In aged mice, it delays skeletal muscle degeneration by increasing the expression of some myokines with potential anti-atrophic activity to levels of young mice (adiponectin, Angptl4, S100a8, and Secreted protein acidic and cysteine rich (Sparc)) [190,191,192]. Interestingly, some of these myokines (S100a8 and Sparc) [193,194] can be also increased by exercise in humans, indicating further similarities in the ways of molecular actions of nutraceuticals and physical activity (Figure 2). In vitro curcumin limited atrophy of PIF- or dexamethasone-treated C2C12 myotubes [195,196,197]; in vivo, it attenuated muscle wasting of MAC16-bearing mice [198]. Conversely, in rats bearing Yoshida AH-130 hepatomas, curcumin, despite its clear antitumoral effects, did not reverse CC [199]. In a retrospective study, patients with advanced pancreatic cancer treated for two months with curcumin lost more weight than controls, losing both fat and muscle [200]. The ongoing clinical trial CurChexia (NCT04208334) will evaluate the effect of curcumin (4000 mg/day × 60 days) against CC in patients with stage III–IV head-and-neck cancer, possibly clarifying the usefulness of curcumin against CC.

Mounting evidence shows that resveratrol, one of the main polyphenols in grapes, prevents muscle atrophy in a large number of catabolic conditions, such as disuse [201], aging [202,203], diabetes [204] and cancer cachexia [205]. It may also be useful to maintain a healthy weight against obesity, by modulating the secretion of many myokines, including irisin [206,207]. Resveratrol stimulates IGF1 signaling pathway, favoring hypertrophy in newly formed myotubes [208]. Some evidence suggests resveratrol may protect against ROS by restoring silent mating type information regulation 2 homolog-1 (SIRT1) levels, resulting in reduced mitochondrial-related apoptosis in myoblasts [209]. SIRT1 also seems to be the target by which resveratrol can blunt dexamethasone-driven catabolic effects in L6 myotubes [210]. Indeed, SIRT1 reduces muscle wasting by blocking the activation of FoxO1 and 3 and, thus, prevents catabolism by hindering the induction of atrogenes [211]. In C26-bearing mice, dietary supplementation with resveratrol inhibits both skeletal and cardiac muscle atrophy by impairing the DNA binding activity of the NF-kB (p65) subunit [212]. Most importantly, it has been recently shown that in a cachectic mouse models of pancreatic cancer, resveratrol is able to restrain muscle mass and strength by reducing the SIRT1-NOX4 signaling in muscle, being NOX4 a key modulator of ROS production [205]. However, intraperitoneal injection of resveratrol did not counteract muscle wasting in mice bearing LLC or Yoshida AH-130 hepatomas [213]. This illustrates how the route of administration and dose may affect bioavailability—and the bioactivity—of this and other nutraceuticals.

Quercetin, found in many fruits, vegetables, and seeds, is another SIRT1 inducer that has given promising results in pre-clinical settings. Quercetin has the potential to reduce OS-induced damage by acting on the TNFα, AKT, PGC1α, and AMPK pathways (reviewed in [214]). In C2C12 cells, quercetin suppressed dexamethasone-induced expression of atrogin-1 and MuRF1, and myostatin too, in a concentration-dependent manner [215]. It also reduced dexamethasone-stimulated ROS production by reversing the mitochondrial membrane potential imbalance and apoptosis by regulating the Bax/Bcl-2 ratio [216]. In vivo, seven days pre-treatment with quercetin glycosides (0.45% *w/v* in drinking water) prevented dexamethasone-induced gastrocnemius atrophy of mice, by downregulating atrogene expression [215]. In Apc^Min/+^ mice, three weeks treatment with 25 mg/kg quercetin (oral gavage), after tumor development, reduced muscle atrophy by lowering plasma IL-6 and muscle STAT3 activation [217]. In the same animal model, daily quercetin given by gavage (75 mg/kg) also attenuated grip strength loss that was abolished by the overexpression of IL-6 via muscle electroporation [218], further indicating that the IL-6/STAT3 pathway is very likely the target of quercetin [217]. C26 tumor-bearing mice fed a quercetin-enriched diet (35 mg/kg/day for 21 days) showed preserved body and hindlimb muscle weights [219], whilst daily intraperitoneal injections of quercetin (10 mg/kg for 15 days) prevented cachexia and increased survival of rats bearing Walker 256 tumor, though this effect may be related to an anticancer function of this flavonol [220].

Thus, a large body of evidence supports the use of polyphenols to halt muscle wasting in experimental models of CC. The different methods used for polyphenol supplementation call for more homogeneity in pre-clinical settings to understand their bioavailability and thus pave the way for large-cohort clinical studies to test the efficacy of these natural compounds against cachexia.

#### 3.2.3. Other Promising Nutraceuticals against Cancer Cachexia

##### Alkaloids

Alkaloids are a group of naturally occurring chemical compounds that mostly contain basic nitrogen atoms and are used in clinical practice, particularly as anticancer agents [221]. Matrine is an alkaloid extracted from *Sophora flavescens,* a plant found widely in Asia with potent activities against various malignancies [222], including ovarian cancer [223], hepatocellular carcinoma [224], and human non-small-cell lung cancer [225]. Intraperitoneally injected matrine (50 mg/kg for 5 or 11 days from the onset of cachexia) in C26-bearing mice lowered serum levels of TNFα and IL-6 and preserved body and gastrocnemius muscle weights [226], downregulating the expression of atrogenes in skeletal muscle [227]. Matrine also restrains C2C12 myotube atrophy and apoptosis induced by different atrophic stimuli (dexamethasone, TNFα and C26-conditioned medium) by activating the AKT/mTOR signaling pathway and inhibiting FoxO3-mediated atrogene expression [227]. Since matrine can influence mitochondrial function and ROS production in diverse cancer cells [228,229,230] and in oxidized LDL-stimulated macrophages [231], it remains to be established whether it can modulate OS also in muscles during CC.

Theophylline (1,3-dimethylxanthine) is a natural component of tea leaves and green coffee beans, with anti-inflammatory effects on cytokines, primarily produced by peripheral blood mononuclear cells [232]. In C2C12 myotubes, it reduces the proteolysis during hyperthermia-induced atrophy, and similarly to matrine, it partially reverses cachexia of rats bearing Yoshida AH-130 hepatoma, by lowering circulating TNFα levels [233]. Further investigation is needed to clarify the possible role of theophylline in other unrelated CC models such as nude mice bearing human renal cancer or LLC-bearing mice [17,70].

Berberine is an isoquinoline alkaloid found in *Berberis* plants, with positive effects on various OS-mediated metabolic syndromes (reviewed in [234]) and in cancers. Berberine inhibits proliferation and induces apoptosis of colorectal cancer cells by impinging on JNK, p38 mitogen-activated protein kinase (MAPK), and NF-kB-based signaling, and modulating mitochondrial-ROS generation [235]. In nude mice bearing a human esophageal cancer cell line (YES-2) and in C26-bearing mice, oral supplementation with *Coptidis rhizoma* (containing the active component berberine) reduced tumor-derived IL-6 and BWL without changing food intake or tumor growth [236,237]. In mice inoculated with human colon cancer HCT116 cells, intraperitoneally injected berberine alleviated intestinal mucosal damage [238], a hallmark of CC that may further exacerbate systemic inflammation and cause food malabsorption, reducing food intake [239].

It is in this context that we recently investigated the anti-cachexia potential of trabectedin, a marine drug approved for the treatment of advanced solid tumors, and its less hepatotoxic analog lurbinectedin [240]. Both drugs greatly extended the lifespan of C26-bearing mice, and though lurbinectedin and trabectedin did not protect C26-bearing mice from muscle wasting, lurbinectedin strikingly reduced NF-kB/PAX7-related muscle inflammation and splenomegaly, without affecting tumor growth [157].

Other natural alkaloids have not been tested for managing cachexia but call for proper investigation. For instance, tomatidine is a natural small molecule found in the stems and leaves of tomato plants and unripe tomatoes. Tomatidine seems able to prevent oxidative damage under diabetic conditions in different tissues (retina, kidney, and skeletal muscles) [241]. It promotes skeletal muscle hypertrophy, by activation of the mTOR complex 1 (mTORC1) anabolic pathway, leading to increased strength and exercise capacity in young healthy mice. Most importantly, it reduces skeletal muscle atrophy during fasting and limb immobilization [242,243]. Therefore, it remains to be seen whether tomatidine, by antagonizing OS, will have a beneficial effect also against CC-mediated muscle atrophy.

##### Triterpenoids

Triterpenoids are a class of metabolites composed of three terpene units, biosynthesized in plants by the cyclization of squalene, a biochemical precursor of all steroids, widely used as chemo-preventive and anticancer agents [244], though their exact mechanisms of action have not been elucidated yet (reviewed in [245]).

Among the most important triterpenoids, ursolic acid, a pentacyclic triterpenoid found in various fruits and vegetables, displays anti-inflammatory, cardioprotective, and antitumoral properties. Ursolic acid exerts anticancer action by restraining various inflammation-related pathways, including STAT3, NF-kB, and MAPK (reviewed in [246]). As suggested by Shen and collaborators, it can also modulate ROS production, by triggering endoplasmic reticulum stress and autophagy via a ROS-dependent pathway, at least in human glioma cells [247]. In muscles, Ebert et al. found that ursolic acid promoted muscle growth through an mTORC1-dependent mechanism and reduced muscle atrophy and weakness in aged mice by repressing atrogene expression and transcription factor ATF4 activation [248]. In a mouse model of chronic kidney disease, with associated muscle wasting, 3 week treatment with ursolic acid (100 mg/kg/day, by oral gavage) stimulated muscle protein synthesis, suppressed muscle protein degradation, and lowered plasma levels of inflammatory cytokines (TGF-β, IL-6, and TNFα) [249]. Similarly, 5 day-treatment with ursolic acid (100 mg/kg/day, by oral gavage) in dexamethasone-treated rats protected them from muscle damage and weakness and reduced MuRF1 expression by limiting the nuclear translocation of FoxO1 [250].

Ginsenoside Rg1 belongs to the class of triterpene saponins and is the major pharmacologically active compound found in *Panax ginseng* [251]. It is one of the most famous traditional medicinal herbs, widely used for many centuries in Asian and Western countries. On account of its anti-inflammatory and antioxidant actions, it has been amply investigated for treating some chronic diseases [252,253,254]. Five week access to 0.4 mg/mL Rg1-containing water enhanced muscle size and function of healthy mice by increasing both the oxidative and glycolytic metabolism in muscle fibers. Ginsenoside Rg1 boosted mitochondrial oxidative metabolism and protein synthesis in muscles, the latter by activating S6K but inhibiting 4E-BP1 [255]. However, Lu et al. reported that C26-bearing mice orally treated with ginsenoside Rg1 (10.72 mg/kg in Phosphate-buffered saline (PBS) solution for 23 days from the onset of cachexia) had neither the protection of body weight nor gastrocnemius muscle mass but only reduced levels of TNFα and IL-6, further confirming that solely inhibition of proinflammatory pathways is not sufficient to block CC [256]. This study may contrast with data reported in [255], suggesting Rg1 as a valuable nutritional supplement for protecting muscles from atrophy. However, Rg1′s lack of efficacy reported by Lu et al. might be explained by the lower cumulative dose used for treating the cachectic mice [256] compared to that given to healthy mice in the study by Jeong and coworkers [255].

Glycyrrhizin, a triterpene saponin extracted from licorice (*Glycyrrhiza glabra*), also claims numerous beneficial effects against chronic diseases (reviewed in [257]). For instance, it has anticancer properties, since it induces ROS-dependent apoptosis and cell cycle arrest, at least in HeLa cells [258]. It also has anti-cachexia effects in a mouse model of lung adenocarcinoma, though this may possibly be due to glycyrrhizin-mediated reduction in both tumor progression and cisplatin toxicity [259]. Furthermore, Ayeka et al. reported that C26-bearing mice orally given different licorice extracts for 14 days displayed decreased levels of TNFα and partial recovery in their body weight at the end of treatment [259]. Surprisingly, even if glycyrrhizin is known to inhibit HMGB1 [260], which seems increased in plasma and muscles of cachectic C26 mice [13] but not in muscles of LLC-bearing ones [261], it is still not clear whether the beneficial effects of this molecule against cachexia are related to the inhibition of HMGB1.

Further studies are needed to fully ascertain whether glycyrrhizin, as well as other triterpenoid compounds, are also useful for tackling CC in other unrelated preclinical models, and to confirm their role as a modulator of OS and/or inflammation during CC, before moving to clinical trials.

## 4. Antioxidant and Anti-Inflammatory Effects of Exercise for Cancer Cachexia

Physical activity has many beneficial effects and promotes overall wellbeing, improving muscle strength and metabolic function [262]. Metabolic adaptations to exercise are specific and depend on the nature of the exercise. Physical activity can be classified as “resistance/strength” or “endurance/aerobic” training. Resistance exercise consists of short periods of high contractile muscle performance against heavy external load, and when performed routinely, it mostly leads to increased muscle mass and strength and increased expression of glycolytic enzymes [263]. Instead, the endurance training consists of prolonged periods of low contractility, which results in acute rise in muscle ATP turnover and increased blood flow. It mainly promotes the shift of muscle fibers towards those relying more on oxidative metabolism, favoring intramuscular energy stores and mainly promoting more resistance to fatigue [264].

The American Cancer Society recommends cancer patients avoid inactivity and “take part in regular physical activity” of at least 150 min per week, including strength training exercises at least two days a week (https://www.cancer.org). Cancer Research UK supports these recommendations and suggests that all adults should try either 150 min of moderate-intensity activity or 75 min of vigorous activity per week (https://www.cancerresearchuk.org). They also state that “all adults should also try and build strength twice a week (e.g., weight training or yoga)” for preventing cancer and other pathologies. As regards these latter, both walking and vigorous exercise reduce the incidence of cardiovascular events among postmenopausal women [265] and strength training prevents sarcopenia (i.e., age-related muscle atrophy) [266].

Regular exercise seems to be a valid non-pharmacological approach to a wide variety of chronic diseases associated with low but chronic systemic inflammation, through amelioration of the inflammatory profile. Regular physical activity can reduce pro-inflammatory cytokines (i.e., TWEAK) [267] and raise the plasma levels of several anti-inflammatory ones, such as IL-10, IL-1 receptor antagonist (IL-1ra), and soluble TNF receptors-1 and -2 (sTNF-r1 and sTNF-r2), by monocytes [268,269,270]. IL-6 and other muscle-derived cytokines, referred to as myokines, are released by muscles in response to physical activity and mediate multiple beneficial metabolic effects on other organs (lipolysis and fat oxidation) [269,271,272,273].

Myostatin (also known as growth differentiation factor 8, GDF-8) is a myokine that negatively affects muscle mass and is reduced by both types of exercise [274,275], further linking inactivity with muscle loss. At the same time, myostatin has been found related to increased muscle wasting during cancer [26,276], representing a possible target against CC. Follistatin by antagonizing myostatin has been shown to exert anti-atrophic effect at least in a specific subset of cancer-bearing mice, the inhibin-deficient ones [277]. Drawing on this evidence, physical exercise could potentially be a promising intervention strategy for the prevention and/or treatment of CC because of its simultaneous ability to boost muscle strength, lower overall fatigue, and reduce systemic inflammation [278]. A single bout of resistance exercise has been found able to decrease Fbxo40 in human muscles, suggesting increased protein synthesis [279], while if and how the expression of atrogin-1 and MuRF1 changes upon different types of exercise has given promiscuous results [279,280,281,282], perhaps depending on the different muscle analyzed.

Many pre-clinical studies in CC animal models have investigated the impact of both types of training on muscle physiology and the molecular signaling pathways involved (Figure 2). Alves and coworkers demonstrated that 16 days of high-intensity interval training, consisting of five intervals of 3 min treadmill running at 18 m/min followed by 4 min running at 25 m/min, counteracted cachexia, prolonged survival, and also delayed tumor progression in LLC-bearing mice [283]. In a previous study, Gui et al. showed that four weeks of treadmill exercise (12 m/min, for 60 min/day, five days/week) did not affect tumor growth [284] but partially rescued muscle weight and reduced atrogin-1 expression in gastrocnemius of LLC-bearing mice. Interestingly, Jee et al. demonstrated that four weeks of intense treadmill exercise (~16m/min, 45 min/day, once every two days) was more efficient in limiting muscle atrophy and improving the quality of life and survival rate compared to moderate treadmill sessions (~8m/min, 45 min/day, once every 2 days) in a lung cancer mouse model of CC [285]. Similarly, eight weeks of moderate treadmill exercise (18 m/min, 1 h, 6 days/week, 5% uphill) attenuated BWL in Apc^Min/+^ mice electroporated with IL-6-containing plasmid in their quadriceps [286]. Treadmill exercise normalized insulin sensitivity and improved the muscle metabolism of Apc^Min/+^ mice, by reducing the activation of AMPK and inducing AKT signaling, through improved muscle oxidative capacity [286]. Rats bearing the Yoshida AH-130 hepatomas and performing eight sessions of low-intensity treadmill exercise (15 m/min, 30 min/session) had preserved muscle mass compared to sedentary counterparts, through suppression of the ubiquitin–proteasome pathway, increased hypoxia-inducible factor 1α (HIF-1α) levels, and phosphorylated AMPK [287].

Voluntary wheel running also preserved muscle mass and forelimb grip force in a transgenic mouse model of prostate adenocarcinoma, by reducing myostatin levels [288]. This type of exercise also extended the survival of C26-bearing mice and reduced their loss of muscle mass and function [289], by restraining atrogene expression and restoring the autophagic flux (i.e., lower LC3bII/LC3bI ratio and p62 levels) [290]. To mimic some of the effects of aerobic exercise, Pigna and coworkers treated C26-bearing mice with either AICAR, a pharmacological activator of AMPK [291], or rapamycin, an mTOR inhibitor [292]. Both drugs fully counteracted atrogene induction and muscle atrophy, by triggering the autophagic flux in Tibialis Anterior muscles of these mice. Similarly, trimetazidine (TMZ) given to C26-bearing mice exerted some exercise-like effects, such as boosting grip strength, fast-to-slow myofiber shift, PGC1α induction with promotion of mitochondrial biogenesis, and protection of myofiber cross-sectional area (CSA) [293]. Coletti and collaborators further found that voluntary wheel running downregulated PAX7 expression and NF-kB activation in muscles from C26-bearing mice, preserving muscle mass and fiber size, thus removing the myogenic differentiation block observed in CC [294].

Furthermore, the role of combined resistance and endurance activity was explored in C26-bearing mice. Climbing a ladder inclined at 85°, with weights tied to their tails, was used for resistance training, while endurance exercise was carried out on the same day using a motorized wheel (5–9 m/min, 25 min/day) [295]. Mice were exercised for 4 days/week, for four weeks before tumor implantation and 11 days after C26 implantation. Both muscle mass and strength were improved by the combined training, resulting from modulation of both autophagy and proteasome-mediated degradation, as in [290]. Surprisingly, combined training did not rescue PGC1α protein levels [295], which are typically induced by endurance training in this mouse model of CC [272,296], and amply reviewed in [67]. On the same line, Ballarò et al. showed that 12 days of a moderate mixed type of exercise on a hill motorized treadmill (11 m/min, 45 min, 3 days out of 4) in C26-bearing mice relieved muscle wasting, prevented loss of muscle strength, and partially reduced muscle OS, by improving the antioxidant capacity [84].

A beneficial effect of exercise on OS in muscle has been confirmed in Walker-256 tumor-bearing rats exposed for six weeks to resistance exercise, climbing a ladder apparatus with weights tied to their tails. This type of exercise mitigated BWL and muscle wasting in these rats by attenuating muscle OS and systemic inflammation [297]. However, Khamoui et al. found that three days per week for 11 weeks of ladder climbing with weights did not prevent C26-induced body and muscle weight loss in mice but induced the expression of genes associated with muscle damage and repair [298]. They found that only aerobic wheel running training partially rescued gastrocnemius mass, possibly by activating the mTOR pathway [298]. All these pre-clinical studies support the beneficial effect of physical exercise for counteracting CC, even if some discrepancies can be ascribed to the variety of cachexia models used, diverse background strains, and different exercise protocols.

Besides preclinical models, endurance as well as resistance training seems to offer therapeutic promise for CC patients too, perhaps, at least in part, because they promote antioxidant activity and anti-inflammatory response [84,299,300,301]. Both training programs, despite their different intensities, length, and frequency, improve muscle functional status, body composition, and promote the survival of patients with malignancies, suffering from cachexia [302,303]. In cancer survivors, on stopping radiation or chemotherapy, an endurance training program can mitigate cancer-related fatigue [304], compared to standard care, but also increases muscle strength, cardiorespiratory fitness, and reduces OS markers [305], once again highlighting the usefulness of chronic aerobic exercise in promoting antioxidant capacity.

Other studies also indicated the advantages of resistance exercise for CC patients. Progressive resistance training improves muscle strength and body weight in patients suffering from pancreatic cancer [306] or prostate cancer under radiation therapy [307] or breast cancers [308]. Resistance training also had beneficial effects in maintaining adequate quality of life in cachectic head-and-neck cancer patients undergoing radiotherapy [309,310]. Overall, it seems that resistance training is useful to improve muscle strength and composition in oncologic patients, while aerobic one to reduce the cardiovascular- and metabolic-related complications of cancer and/or concomitant therapy [311].

Despite all this evidence and the recommendations of the American Cancer Society and Cancer Research UK, there is no consensus on the best modality of exercise training for cancer patients. From a general standpoint, since glycolytic fibers (primarily recruited by high-intensity strength training [312]) are more susceptible than oxidative myofibers to tumor-induced atrophy [14], it may seem logical to recommend that cancer patients at risk of developing CC do more endurance exercises to stimulate oxidative metabolic pathways and, thus, to target these specific fibers. Strenuous physical exercise is not feasible for patients who are more fragile because of aggressive therapies. Cancer patients should take moderate endurance/aerobic physical activity to potentiate muscle capacity and to detoxify ROS, through non-enzymatic and enzymatic antioxidant responses (discussed in [299]). An example of moderate physical activity includes three to five simple exercises, such as sit-to-stand, knee extension, or similar, repeated three times a day at home [313].

However, this kind of training has beneficial effects when endurance activity is constant and progressive, which is hard for CC patients who often suffer from chronic fatigue, anemia [314], and cardiovascular impairment, limiting their exercise capacity. Therefore, it is important to develop methods to enhance the effects of even low/moderate intensity exercise.

## 5. Combination of Physical Activity and Nutraceuticals against Cancer Cachexia

Various studies support the use of nutraceuticals (e.g., PUFAs, ursolic acid, cocoa flavanols, ginsenoside, and curcumin) not only for boosting physical performance—by rising the serum levels of some beneficial myokines too, such as irisin [315]—but also for preventing/reducing OS and, consequently, inflammation and muscle damage, elicited by intense physical training [316,317,318,319,320,321]. At least in the case of sarcopenia, the nutraceutical/exercise combination has already given promising results in aged subjects by improving their strength and physical performance and by reducing OS and muscle fatigue (reviewed in [322]). A few in vivo studies have investigated a multimodal approach as a feasible beneficial strategy for counteracting CC. For example, Penna and coworkers found that EPA supplementation of LLC-bearing mice subjected to treadmill (14 m/min, 45min, 5 days/week) partially rescued muscle strength and mass by raising PGC1α levels and favoring muscle regeneration [323]. Of interest, resveratrol promoted muscle regeneration in C26-bearing mice exercised through 6 weeks of resistance training, climbing a ladder with weights tied to their tails [324].

Despite the scarce pre-clinical findings, some clinical trials have already evaluated this bimodal approach. A summary of them is included in Table 1. In 2017, a phase II trial with only 46 participants confirmed the feasibility and safety of a six-week multimodal intervention, which included PUFA supplementation, aerobic and resistance exercise, and anti-inflammatory therapy (celecoxib) for CC patients with advanced non-small-cell lung cancer (stage III–IV) or pancreatic cancer not eligible for curative therapy [325]. Based on the positive effects observed both on body weights and skeletal muscle mass but not on overall performance, phase III is ongoing under the name EudraCT 2013-002282-19 to definitely clarify the usefulness of this combined approach to CC.

To extend these findings, the ongoing multimodality exercise/nutrition anti-inflammatory treatment for cachexia (MENAC) phase III trial is currently enrolling 240 patients with diagnosis of lung or pancreatic cancer or cholangiocarcinoma. The intervention arm consists of six weeks of nutritional counseling, oral nutrition supplementation (including EPA), a mixed physical exercise program and ibuprofen, compared to standard care, for early prevention of cachexia [326]. Similarly, a recent single-arm intervention study assessed the feasibility of a multimodal intervention (2 g EPA/DHA via fish oil daily, regular dietary counseling, and unsupervised mixed physical exercise twice weekly) for CC management in 59 patients with non-small-cell lung cancer during the first three cycles of primary anti-neoplastic treatment. The gain in skeletal muscle mass was greater in patients receiving this multimodal intervention than in patients receiving standard care alone, though there were no differences in mean skeletal muscle, body weight, or physical function [327]. Adherence to the treatment plan and compliance with the intervention were among the contributing factors that favored the muscle gain in cancer patients [327].

Unfortunately, poor compliance in these trials, mostly due to deterioration of both psychological and health status [328], especially in completing physical exercise programs [325,327,329], may explain the scarce efficacy in ameliorating physical function and preventing body and muscle wasting in CC patients. The inclusion of advanced cachectic patients may further limit the feasibility of these promising therapies because of the high risk of dropouts. It would therefore be desirable to find other interventions, more easily to be completed.

A pilot study led by Schink and coworkers evaluated the effect of whole-body electromyostimulation (WB-EMS) to simultaneously stimulate muscles of the upper legs, arms, the bottom, the abdomen, the chest, and the back in advanced cancer patients under anticancer treatment. Twenty minutes of WB-EMS twice a week for 12 weeks combined with a very easy supervised exercise program and individualized nutritional support were more effective in sustaining skeletal muscle mass and improving physical function than standard dietary therapy alone [330]. An ongoing clinical trial (NCT03151291) is testing the combined approach of WB-EMS and specific nutritional support (including EPA) in patients with solid or hematological cancers. This study might pave the way to future large-scale trials for testing the feasibility and efficacy of this new exercise scheme, combined with appropriate nutraceuticals to improve not only body composition and muscle function but also oxidative/inflammatory status and, ultimately, the quality of life of CC patients.

## 6. Conclusions and Prospects

In summary, current evidence suggests that nutraceuticals have the potential to limit CC, although optimal therapy may depend on several factors, especially bioavailability. For instance, new nutraceutical formulations are needed for curcumin supplementation [331], possibly using nanoparticles or liposomes, to increase the bioavailability and subsequently the clinical efficacy of the nutraceutical [331].

Strategies aimed at fortifying the gut barrier function in CC patients—a factor that limits their capacity to absorb both vital nutrients and also nutraceuticals [332]—should be pursued. On the other hand, some standard therapeutic strategies are based on ROS-generating drugs to induce cancer cell apoptosis, so antioxidant nutraceuticals in supra-physiological doses may be detrimental, since they may favor tumor growth and subsequently CC progression. Similarly, combinations of antioxidants may be trivial and counterproductive as already shown in pre-clinical CC models [85]. Therefore, further investigation of patients’ redox status, before starting and during treatment may be useful not only for staging CC patients but also for planning early and appropriate target medication, which may include nutraceuticals. For instance, measuring biomarkers of OS (such as F_2_-isoprostanes and 8-hydroxydeoxyguanosine) in body fluids (e.g., urine) may be rapid and feasible for constant monitoring of OS status in these patients [333].

Regarding physical exercise, endurance/aerobic training may be more beneficial than resistance training in counteracting muscle atrophy and improving the redox status during CC. However, the poor compliance of CC patients performing regular exercise is a major limitation. Thus, nutraceuticals may be a valid support against CC, especially in advanced stages, to potentiate the beneficial effect of a more moderate and attainable exercise program. Nevertheless, in this context too, finely tuned redox status regulation during exercise is crucial for muscle homeostasis, and an excess of antioxidant nutraceuticals may be detrimental due to alterations of the signaling pathways involved in muscle adaptation to physical exercise (reviewed in [334]).

WB-EMS may be useful to increase CC patients’ adherence to an exercise plan, but more effort, especially in pre-clinical practice, is needed for a better understanding of whether and how physical exercise and nutraceuticals can act in synergy and/or antagonism to promote the molecular mechanisms against CC-related atrophy. Filling the existing scientific gaps may be helpful to design larger and more homogeneous clinical trials to test the efficacy of this bimodal approach in CC.

## Figures and Tables

**Figure 1 cells-09-02536-f001:**
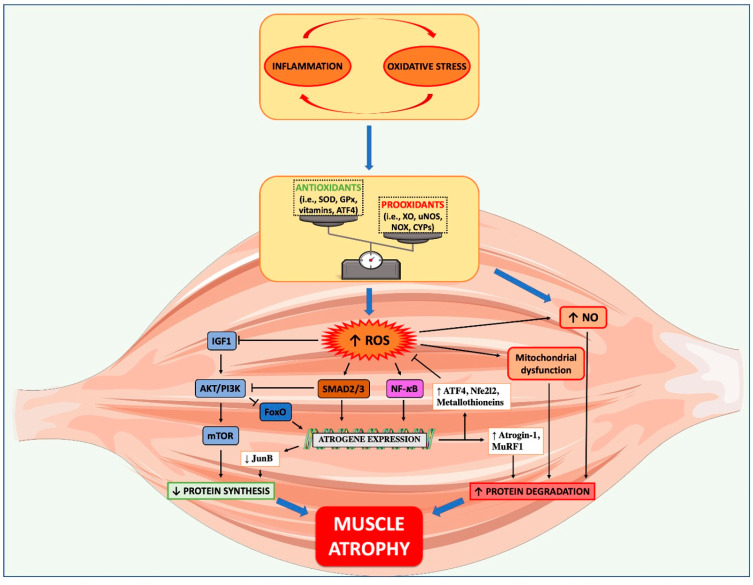
Role of oxidative stress in muscle wasting during cancer cachexia. The crosstalk between increased inflammation and oxidative stress, occurring during cancer cachexia (CC), can drive in muscles an imbalance between antioxidant and pro-oxidant systems, favoring the generation of different sources of reactive oxygen species (ROS). During CC, ROS can reduce protein synthesis, by inhibiting IGF1/PI3K/mTOR pathway, and increase protein degradation, by stimulating pro-atrophic pathways such as FoxO3, SMAD2/3, and NF-κB. The latter promotes the ubiquitin proteasome system activation (atrogin-1 and MuRF1) that, along with ROS-mediated nitric oxide (NO) accumulation and mitochondrial dysfunction, facilitates the hypercatabolism typical of CC. Despite the concomitant activation of detoxifying systems (ATF4, Nfe2l2, metallothioneins), OS seems not to be resolved in muscles during CC.

**Figure 2 cells-09-02536-f002:**
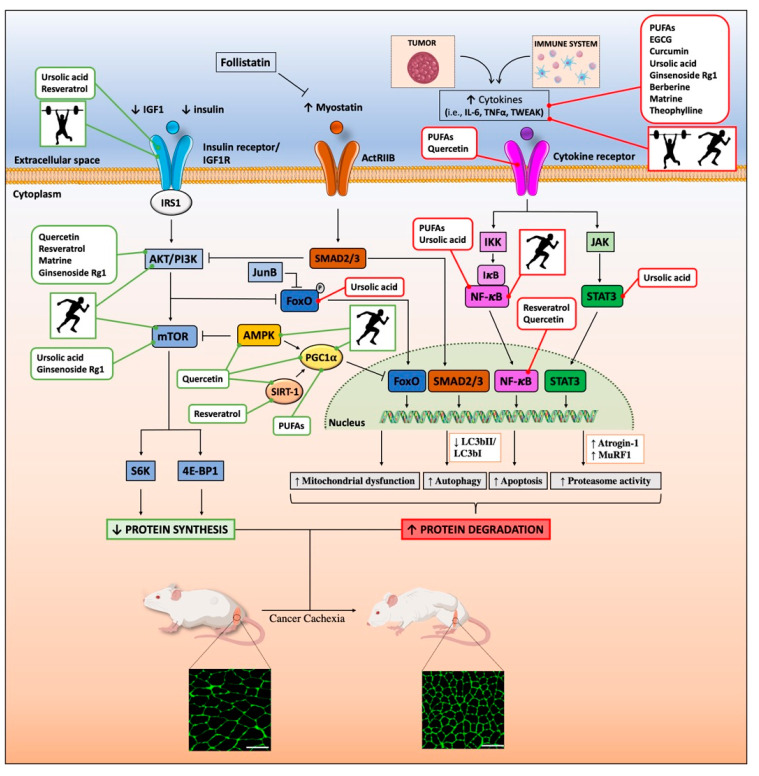
Beneficial effects of nutraceuticals and physical activity against muscle atrophy during cancer cachexia. Graphical representation of the main findings of in vitro and in vivo studies, which have investigated the mechanisms underlying the benefits of the nutraceuticals and physical exercise against muscle wasting during cancer cachexia. In brief, nutraceuticals and physical exercise can favor protein synthesis in muscle by inducing both anabolic (i.e., IGF-1/AKT/mTOR) and anti-catabolic pathways (i.e., AMP-activated protein kinase (AMPK), SIRT1, and PGC1α) (in green). On the other hand, they can limit protein degradation by inhibiting pro-catabolic pathways (i.e., FoxO3, SMAD2/3, and NF-κB) (in red), thus limiting the expression of genes involved in mitochondrial dysfunction, autophagy, apoptosis, proteasomal degradation, and inflammation. The running man and the weightlifter represent “endurance/aerobic” and “resistance/strength” trainings, respectively. A transversal section of a tibialis anterior stained for wheat germ agglutinin (WGA) and coming from a healthy and a colon adenocarcinoma 26 (C26)-bearing mouse are shown. Scale bar: 100 μm.

**Table 1 cells-09-02536-t001:** Summary of human studies showing possible beneficial effects of nutraceutical or physical exercise, alone or combined, against cancer cachexia.

Nutraceutical/Exercise	Study Name	Study Type	Number of Participants	Duration (months)	Intervention	Summary of Findings	References
**n-3 PUFA**	Enteral omega 3 during radiotherapy to improve the quality of life and functionality of head and neck cancer patients	Randomized double-blinded, controlled clinical trial	86	39	Supplementation of an omega-3 highly concentrated substance (5 mL/day, containing 2.25 g of EPA and 1.08 g of DHA) to the standard enteral diet during radiotherapy	Ongoing	NCT03720158
**Curcumin**	The effects of curcumin (diferuloylmethane) on body composition of patients with advanced pancreatic cancer	Retrospective matched 1:2 case-control study	66	15	Supplementation of curcumin (8000 mg/day) for 60 days	Reduced body weight, fat and muscle tissues	[200]
	The effect of curcumin for treatment of cancer anorexia-cachexia syndrome in patients with stage III-IV of head and neck cancer (CurChexia)	Phase II, randomized and double-blinded clinical trial	96	14	Supplementation of curcumin (4000 mg/day) for 60 days	Ongoing	NCT04208334
**Resistance training (RT)**	Physical exercise for patients who suffer from weight loss due to head and neck cancer undergoing medical treatment	Pilot study,2-arm, randomized controlled trial	20	26	Progressive RT (PRT) (30 min, 3 times/week) for 6 weeks, during the radiotherapy	The training program is safe and feasible Improved general fatigue and quality of life	[309]
	PRT rebuilds lean body mass in head and neck cancer patients after radiotherapy: DAHANCA 25B trial	Multi-center, randomized, stratified and parallel-grouped study	41	19	Two group of patients with completed anti-neoplastic therapy: ‑ Early Exercise: initiated 12 weeks of PRT followed by 12 weeks of self-chosen physical activity ‑ Delayed Exercise: initiated 12 weeks of self-chosen physical activity followed by 12 weeks of PRT	Increased lean body mass and muscle strength in both groups	[310]
	Exercise intervention study for pancreatic cancer patients (SUPPORT)	Randomized controlled intervention trial	65	38	Supervised RT or home-based RT (2 times/week) for 6 months during and after chemotherapy	Improved muscle strength supervised RT is more effective than home-based RT	[306]
**Resistance and aerobic training**	Oxidative stress and fitness changes in cancer patients after exercise training	Pilot study, pseudo-randomized trial	22	3	Supervised resistance and aerobic training (60 min/day) for 10 weeks, after 6 weeks of completing anti-neoplastic treatment	Increased muscular strength, antioxidant capacity Decreased markers of protein and DNA oxidation	[305]
**Multimodal intervention**	A feasibility study of multimodal exercise/nutrition/ anti-inflammatory treatment for cachexia (the Pre-MENAC)	Phase II, randomized and open-label feasibility trial	46	38	Celecoxib (300 mg/day) + EPA (2 g/day) + home-based aerobic (30 min, 2 times/week) and RT (20 min, 3 times/week) for 6 weeks during the chemotherapy	A multimodal intervention is feasible and safe No significant effect on physical performance or muscle mass	[325]
	Multimodal intervention for cachexia in advanced cancer patients undergoing chemotherapy (MENAC)	Phase III, randomized controlled trial	240	69	Ibuprofen (1200 mg/day) + EPA (2 g/day) + DHA (1 g/day) + RT (3 times/week) and aerobic training (2 times/week) for 6 weeks during the chemotherapy	Ongoing	[326]
	Feasibility of a multimodal intervention on malnutrition in patients with lung cancer during primary anti-neoplastic treatment	Single arm non-randomized trial	58	15	EPA/DHA (2 g/day) via fish oil daily + resistance and aerobic training (2 times/week) during the first three cycles of anti-neoplastic treatment	Gain in skeletal muscle mass. No effect on mean skeletal muscle, body weight or physical function	[327]
	Feasibility of early multimodal interventions for elderly patients with advanced pancreatic and non-small-cell lung cancer (NEXTAC study)	Multicenter prospective randomized single arm, phase II study	30	24	Branched-chain aminoacids (2.5 g/day), coenzyme Q10 (30 mg/day), and L-carnitine (50 mg/day) + supervised home-based low intensity RT (20–30 min/day) for 8 weeks during the chemotherapy	Multimodal intervention shows excellent compliance and safety	[313]
**Whole-body electro-myostimulation (WB-EMS) + individualized nutritional support ± nutraceuticals**	Effects of WB-EMS combined with individualized nutritional support on body composition in patients with advanced cancer: a controlled pilot trial	Non-randomized controlled pilot trial	131	60	WB-EMS training (2 times/week, 20 min) + individualized nutritional support (dietary advices: daily protein intake > 1 g/kg bodyweight) for 12 weeks during anti-neoplastic treatment	Supervised WB-EMS training is safe WB-EMS, combined with nutritional support, shows promising effects against muscle wasting and on physical function	[330]
	Effects of WB-EMS and Specific Dietary Supplements on Cancer Patients	Randomized with parallel assignment	200	29	WB-EMS training (2 times/week, 20 min) + individualized nutritional support ± supplementation of HMB (3 g/day) or LC (4 g/day) or EPA (2.2 g/day) for 12 weeks during curative or palliative anti-neoplastic treatment	Ongoing	NCT03151291

Abbreviations: EPA: eicosapentaenoic acid; DHA: docosahexaenoic acid; PUFA: polyunsaturated fatty acid; RT: resistance training; PRT: progressive resistance training; WB-EMS: whole-body electromyostimulation; HMB: β-hydroxy-β-methylbutyrate; LC: L-carnitine.

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
