# Peer review of "Nutraceuticals and Exercise against Muscle Wasting during Cancer Cachexia"

_cells, 2020, doi:10.3390/cells9122536_

Round 1
Reviewer 1 Report
Thank you for giving me the opportunity to review this very interesting manuscript about Nutraceuticals and Exercise Against Muscle Wasting During Cancer Cachexia.
The review focuses on the most relevant study results in this regard. The paper is easy to follow and does have a logical flow. It is very well written and focuses on an interesting topic that is of high clinical relevance. The title and abstract cover the main aspects of the work.
I agree with the authors that in clinical practice, cancer cachexia is a challenging symptom. The author did perform an extensive literature research and the manuscript covers the main aspects of this topic. The review provides an excellent overview overview.
Nevertheless, I would like to address some minor concerns to help to improve the quality of the manuscript.
Abstract:
The authors state “no therapy is available against cachexia“. I would recommend to rather state “no causal therapy is available against cachexia“.
Manuscript:
Could the authors please provide a statement about Anamorelin.
Could the authors please provide a statement about Computed tomography (CT) assessment, as it is the gold standard method of body composition analysis and diagnosis of abnormal body composition phenotypes. CT analysis is particularly useful in cachexia, as CT scans are routinely performed during diagnosis and staging of the disease.
Could the authors please provide a statement about Bioelectrical impedance analysis (BIA) as a commonly used method for estimating body composition, in particular body fat and muscle mass. impedance analysis.
Author Response
REVIEWER 1:
Thank you for giving me the opportunity to review this very interesting manuscript about Nutraceuticals and Exercise Against Muscle Wasting During Cancer Cachexia.The review focuses on the most relevant study results in this regard. The paper is easy to follow and does have a logical flow. It is very well written and focuses on an interesting topic that is of high clinical relevance. The title and abstract cover the main aspects of the work.I agree with the authors that in clinical practice, cancer cachexia is a challenging symptom. The author did perform an extensive literature research and the manuscript covers the main aspects of this topic. The review provides an excellent overview overview.
Reply:
We are glad that the reviewer found our review interesting, complete and well-written.
Nevertheless, I would like to address some minor concerns to help to improve the quality of the manuscript.
Abstract:The authors state “no therapy is available against cachexia“. I would recommend to rather state “no causal therapy is available against cachexia“.
Reply:
We thank the reviewer to help us to implement the review and we have inserted his/her suggestion accordingly.
Manuscript:Could the authors please provide a statement about Anamorelin.
Reply:
We agree with the reviewer and added a sentence quoting anamorelin on page 7 of the improved version of the review.
Could the authors please provide a statement about Computed tomography (CT) assessment, as it is the gold standard method of body composition analysis and diagnosis of abnormal body composition phenotypes. CT analysis is particularly useful in cachexia, as CT scans are routinely performed during diagnosis and staging of the disease.
Reply:
We fully agree with the reviewer and added a sentence as requested at the beginning of page 2 of the improved version of the review.
Could the authors please provide a statement about Bioelectrical impedance analysis (BIA) as a commonly used method for estimating body composition, in particular body fat and muscle mass. impedance analysis.
Reply:
We fully agree with the reviewer and added a sentence as requested at the beginning of page 2 of the improved version of the review.
All the changes are highlightened in yellow in the text.
Reviewer 2 Report
A very well written and valuable review of the state of the art in treating the cachexia that is a detrimental and morbid accompaniment to many cancers. The few corrections or suggestions that I have are in the copy I am appending. Brilliant illustrations.

Author Response
REVIEWER 2:
Comments:
A very well written and valuable review of the state of the art in treating the cachexia that is a detrimental and morbid accompaniment to many cancers. The few corrections or suggestions that I have are in the copy I am appending. Brilliant illustrations.
Reply:
We are happy to know that the reviewer appreciated our review and the included illlustrations. We have improved our manuscript by taking advantages of his/her appended suggestions.
All the changes that we made are highlightened in yellow in the text.
Reviewer 3 Report
In the present manuscript, authors have reviewed the potential application of bioactive compounds/nutraceuticals in the management of cancer-associated cachexia syndrome, which accounts for a significant proportion of cancer-related deaths. The authors have provided the appropriate introduction of metabolic syndrome along with a brief molecular mechanism of wasting. Further, they have reviewed the preclinical studies presenting anti-cachectic properties of PUFA, natural polyphenols such as EGCG, curcumin, etc. The authors should include evidence of the anti-cachectic effect of Silibinin (PMID: 26510913 ). Although they have mentioned the role of resveratrol, a recent study describing the molecular mechanism of resveratrol mediated SIRT1 activation has been published (PMID: 32441762), these should be included. Furthermore, the authors have described the combinatorial effect of exercise and nutraceuticals in the management of cachexia, that's really interesting. Although the review is already very lengthy, it will be really good if the authors can include one illustration regarding the mode of action of key nutraceuticals. With these minor corrections, I would highly recommend the publication of this review. One more, suggestion, it will be good if authors can increase the text size inside the illustration.
Author Response
REVIEWER 3:
Comments:
In the present manuscript, authors have reviewed the potential application of bioactive compounds/nutraceuticals in the management of cancer-associated cachexia syndrome, which accounts for a significant proportion of cancer-related deaths. The authors have provided the appropriate introduction of metabolic syndrome along with a brief molecular mechanism of wasting. Further, they have reviewed the preclinical studies presenting anti-cachectic properties of PUFA, natural polyphenols such as EGCG, curcumin, etc. The authors should include evidence of the anti-cachectic effect of Silibinin (PMID: 26510913 ). Although they have mentioned the role of resveratrol, a recent study describing the molecular mechanism of resveratrol mediated SIRT1 activation has been published (PMID: 32441762), these should be included. Furthermore, the authors have described the combinatorial effect of exercise and nutraceuticals in the management of cachexia, that's really interesting. Although the review is already very lengthy, it will be really good if the authors can include one illustration regarding the mode of action of key nutraceuticals. With these minor corrections, I would highly recommend the publication of this review. One more, suggestion, it will be good if authors can increase the text size inside the illustration.
Reply:
We are glad to read that also this expert as well as the other two liked our review.
We have mentioned in the text only one of the two papers s/he suggested, because the one about Silibinin having both anticancer and anticachexia properties and not belonging to any of the nutraceuticals listed is out of the topic of our review. Instead, we agree that quoting 32441762 would really add more insights on the resveratrol-related mechanism against muscle loss in cancer cachexia and we are grateful to the reviewer for this precious suggestion (see page 12 of the improved version of the manuscript).
With regard to figure implementation, we believe that figure 2 already includes mode of action of key nutraceuticals. We have increased the text size of the figures as s/he asked and we agree with the reviewer that they look like much better in this shape. Even if not requested, we have changed the position of the table appended such in a way to render it more readable than before.
All the changes have been highlightened in yellow in the text.